# Amoeba-like self-oscillating polymeric fluids with autonomous sol-gel transition

Michika Onoda[1], Takeshi Ueki[2], Ryota Tamate[1], Mitsuhiro Shibayama[3] & Ryo Yoshida[1]

In the field of polymer science, many kinds of polymeric material systems that show a sol-gel transition have been created. However, most systems are unidirectional stimuli-responsive systems that require physical signals such as a change in temperature. Here, we report on the design of a block copolymer solution that undergoes autonomous and periodic sol-gel transition under constant conditions without any on–off switching through external stimuli. The amplitude of this self-oscillation of the viscosity is about 2,000 mPa s. We also demonstrate an intermittent forward motion of a droplet of the polymer solution synchronized with the autonomous sol-gel transition. This polymer solution bears the potential to become the base for a type of slime-like soft robot that can transform its shape kaleidoscopically and move autonomously, which is associated with the living amoeba that moves forward by a repeated sol-gel transition.

[1] Department of Materials Engineering, School of Engineering, The University of Tokyo, 7-3-1 Hongo, Bunkyo-ku, Tokyo 113-8656, Japan. [2] National Institute for Materials Science, 1-1 Namiki, Tsukuba, Ibaraki 305-0044, Japan. [3] Institute for Solid State Physics, The University of Tokyo, Kashiwanoha, Kashiwa, Chiba 277-8581, Japan. Correspondence and requests for materials should be addressed to T.U. (email: UEKI.Takeshi@nims.go.jp) or to R.Y. (email: ryo@cross.t.u-tokyo.ac.jp).

In living organisms, many kinds of dynamic behaviours emerge from the microscopic molecular scale up to macroscopic bulk sizes through hierarchical structures and cooperative interactions. Among these, stimuli-responsiveness of molecules constitutes an important feature of building blocks. This way, physical or chemical signals, such as other specific molecules, trigger for instance structural changes in molecular conformation or self-assembly of molecules, which eventually results in the expression of a specific function. Aside from stimuli-responsiveness, another important feature is autonomy. In living systems, biochemical reactions fuelled by energy result in the spontaneous creation of a dissipative structure far from the equilibrium state with a periodic rhythm, either as a temporal, spatial or spatio-temporal structure. In protists, for example, the spontaneous assembly and disassembly of actin networks plays a central role in muscle contraction, cell division and motility[1]. Even though these concepts are widely accepted as well-known features in biological systems, in most cases, research on designing bioinspired or biomimetic materials is either related to the creation of self-assemblies of molecules as static structures or to expressing stimuli-responsive functions as relaxation process towards the thermodynamically stable equilibrium state. Likewise, the design of living-like materials with autonomous functions far from equilibrium is still a largely unexplored field of research.

Within this context, we have developed synthetic polymeric material systems with autonomous functions. In these systems, a polymer undergoes self-oscillation under constant conditions without any on–off switching triggered by external stimuli, thus resulting in a dissipative structure far from equilibrium. In 1996, we succeeded in the creation of a self-oscillating polymer gel that spontaneously repeated cyclic swelling–deswelling changes resembling a beating heart[2]. The self-oscillating gels were designed by utilizing the Belousov–Zhabotinsky (BZ) reaction, which is a well-known chemical oscillatory reaction that results in periodic redox changes and the emergence of spontaneous spatio-temporal patterns. Since this pioneering work, we have systematically studied these self-oscillating polymers and gels from nanoscale to bulk range sizes[3,4]. In doing so, the potential of these polymer systems for applications, for example, as biomimetic actuators, mass transport systems and functional fluids was explored. For example, it was demonstrated that an object could be autonomously transported in a tubular self-oscillating gel by a peristaltic pumping motion similar to the working principle of an intestine[5]. Further, a self-oscillating polymer brush surface was prepared as artificial cilia implementing autonomous nanoactuation[6]. Moreover, autonomous viscosity oscillations of a fluid were realized by utilizing microgels[7], a metallo-supramolecular complex[8] or block copolymer solutions.[9] In addition, self-oscillation between unimer/micelle[10] or unimer/vesicle (polymersome)[11,12] structures was also achieved by utilizing synthetic block copolymers[7,8]. In particular, an oscillatory shape deformation, which is often observed for living cells during, for example, cell migration or morphogenesis, was realized for vesicles. Furthermore, cell-like hollow spheres composed of self-oscillating microgels (that is, colloidosomes) were fabricated[13], and it was shown that they exhibited drastic shape oscillation as well as swelling/deswelling oscillations. These studies gave rise to the concepts of dynamic functional polymer gels and greatly expanded their potentials. This way, self-oscillating polymers and gels have attracted much attention in many fields of research and have inspired numerous related studies.

In one of our previous studies, we achieved viscosity oscillation with a maximum amplitude of about 2 mPa s (ref. 9). In contrast, here, we have succeeded in increasing this amplitude to 2,000 mPa s, which is equal to the oscillation amplitude in

living amoeba. At the same time, in this present study, a self-oscillating sol-gel transition has been realized. To achieve the significant increase in the amplitude of oscillation, we designed a ABC triblock copolymer that is responsible for drastic structural changes. As a result, this copolymer shows a sol-gel transition with association and dissociation of percolated network structures similar to that in a living amoeba under constant condition without any external stimuli.

## Results

**Design strategy of the sol-gel oscillating polymeric fluids.** The target multiblock copolymer was synthesized by sequential reversible addition fragmentation chain transfer (RAFT) polymerization incorporating a thermoresponsive A segment, a hydrophilic B segment, and a self-oscillating C segment (Fig. 1a).

As the temperature increases, the ABC triblock copolymer undergoes structural changes from unimer to micelle structures, and from their transitions to micelle-connected networks (Fig. 1b). It is well-known from network percolation theory that the elastic modulus is proportional to the number density of the crosslinking point in space. We straightforwardly aim to invoke periodic changes of the effective polymer junction by totally synthetic materials leading to viscoelastic oscillation. The thermoresponsive A segment is composed of $N$-isopropylacrylamide (NIPAAm) and $n$-butylacrylate (BA) (NIPAAm/BA = 85/15 mol%). It showed a lower critical solution temperature (LCST)-type phase transition ($T_A$) owing to the LCST nature of the NIPAAm moiety and the hydrophobicity of BA. In contrast, the self-oscillating C segment was designed to have a LCST-type aggregation temperature ($T_C$) higher than $T_A$ to achieve a stepwise aggregation. This stepwise aggregation can suppress the formation of a loop chain[14–17]. As a result, the efficient formation of a network structure could be achieved. Here, the redox state of Ru(bpy)$_3$ introduced in the C segment affected $T_C$. $T_C$ in the oxidized state ($T_{C, Ox}$) increased more than it did in the reduced state ($T_{C, Red}$) due to an increase in hydrophilicity of the C segment. Therefore, a bistable temperature ($T_b$) emerges between $T_{C, Red}$ and $T_{C, Ox}$. At $T_b$ (region in Fig. 1b with yellow background), an autonomous micellar connection/dispersion oscillation coupled with a redox change of Ru(bpy)$_3$ driven by the BZ reaction could occur without applying any external stimuli. Consequently, in a concentrated polymer solution, autonomous viscosity oscillation and sol-gel oscillation could be realized through percolation of the micellar network structure (Fig. 1c).

**Rheological property of the block copolymer in equilibrium.** First, the viscosity as a function of temperature was investigated in both redox states for 5.0 wt% polymer solution (Fig. 2a). The viscosity abruptly increased above $T_C$ in both the redox states. $T_{C, Ox}$ was found to be 13.9 °C higher than $T_{C, Red}$. Figure 2b shows the temperature dependence of storage ($G'$) and loss ($G''$) moduli for the 5.0 wt% polymer solution. Dramatic rise in $G'$ and $G''$ with transition from sol to gel (that is, $G' > G''$) was clearly observed in the reduced state, whereas only a relatively mild increase was observed in the oxidized state due to the instability of the oxidized state in equilibrium. Importantly, $T_{gel, Ox}$ was 2.2 °C higher than $T_{gel, Red}$. Viscosity, $G'$ and $G''$ of the reduced state were higher than those of the oxidized state in all the temperature region. Second, dynamic light scattering (DLS) measurements were conducted to evaluate the ergodicity of the 5.0 wt% polymer solution (Fig. 2c). The initial amplitude of the intensity correlation function ($\sigma_I^2$) was dramatically decreased at higher temperatures only in the reduced state, which implied that the ergodicity of the solution was shifted from ergodic sol to non-ergodic gel[18]. These results strongly support the possibility to

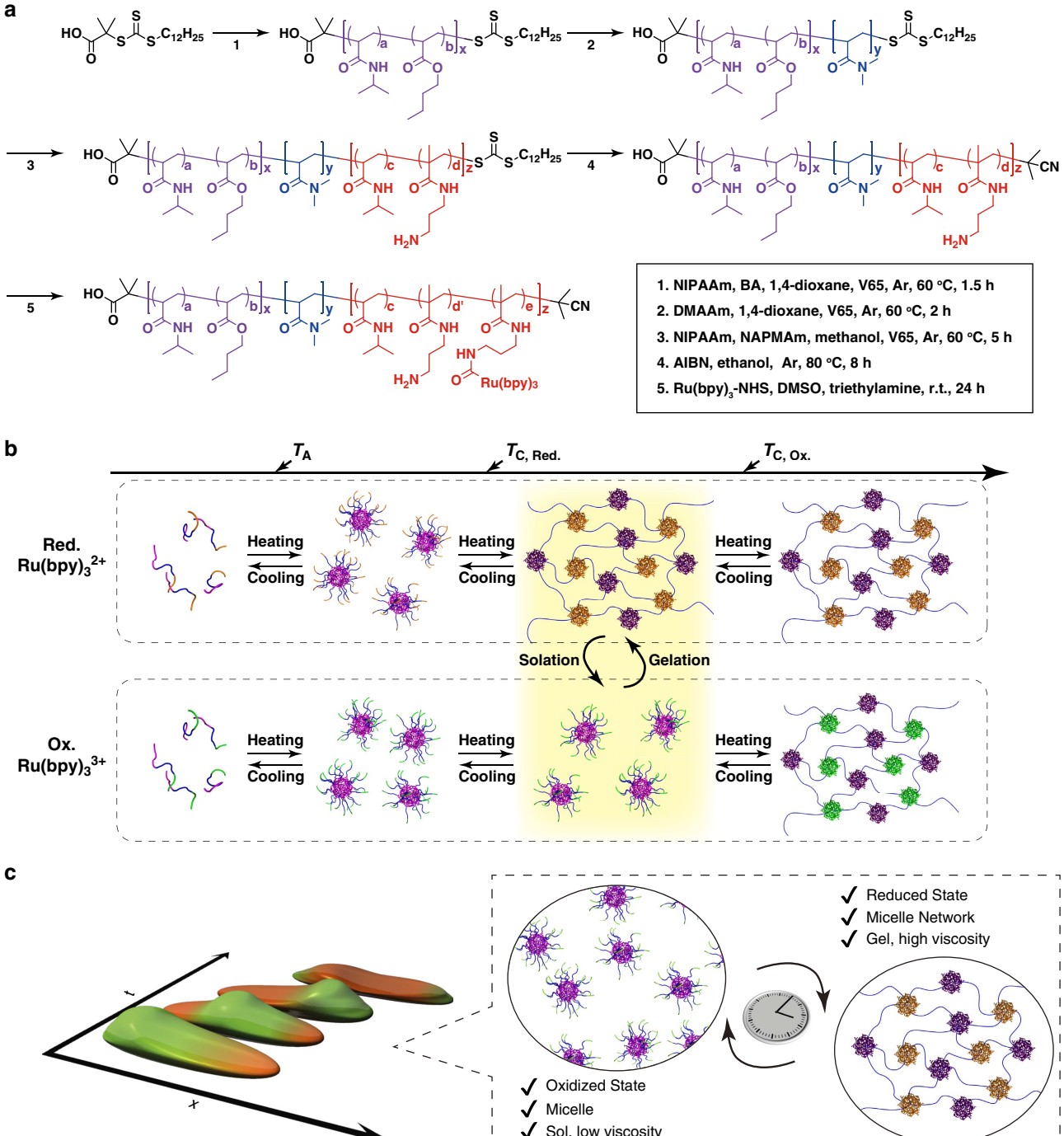

**Figure 1 | Design strategy and conceptual illustration of this study.** (**a**) Synthetic procedure for the ABC triblock copolymer. (**b**) Structural changes from unimer to micelle-connected networks of the ABC triblock copolymer with increasing temperature. (**c**) Conceptual illustration of the autonomous and periodic sol-gel transition based on structural changes driven by redox changes of the ABC triblock copolymer.

achieve autonomous sol-gel oscillation at $T_b$ with appropriate concentration of the BZ substrates.

**Microscopic structural oscillation of the block copolymer.** Further, we conducted preliminary time-resolved DLS measurements for dilute polymer solution (concentration = 0.1 wt%) as a series of investigations of the oscillating properties of the ABC triblock copolymer coupled with the BZ reaction. Supplementary Fig. 4 shows the time-averaged scattering

intensity ($<I>_{nom}$) and the hydrodynamic radius, $R_h$, as a function of time with appropriate concentration of BZ substrates at a temperature of 26 °C. During the BZ reaction, the bottom and top peaks of $R_h$ in oxidized and reduced state were 30 and 60 nm, respectively. This observation suggests that connection/dispersion oscillations of polymeric micelles were established successfully. Furthermore, by increasing the polymer concentration to 0.5 wt%, the top peak in the reduced state rose to around 160 nm, whereas the bottom peak in the oxidized state remained at 30 nm (Supplementary Fig. 5b). Interestingly, in the case of these ABC

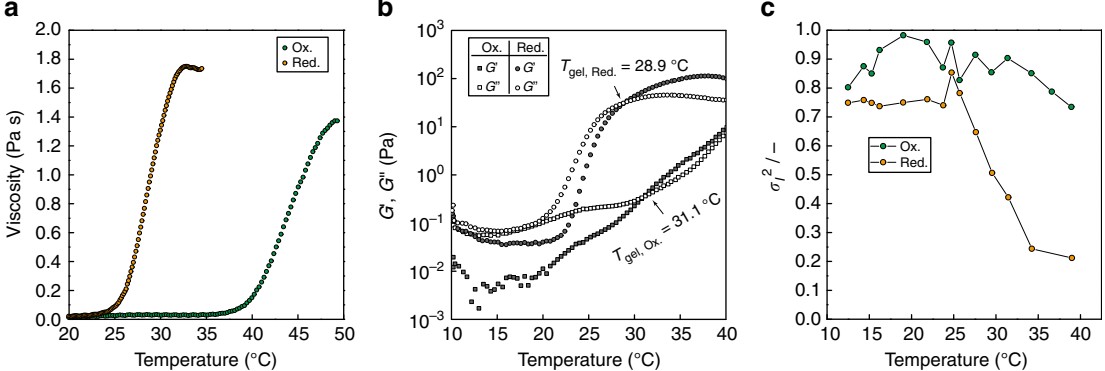

**Figure 2 | Rheological properties of the ABC triblock copolymers under equilibrium.** Temperature dependence of (**a**) viscosity (**1**), (**b**) storage ($G'$) and loss ($G''$) moduli (**2**), and (**c**) initial amplitude of the intensity correlation function, $\sigma_I^2$ (**3**) for reduced and oxidized ABC triblock copolymers (5.0 wt%). The reduced Ru(bpy)$_3^{2+}$ and the oxidized Ru(bpy)$_3^{3+}$ states were achieved by adding 0.81 M HNO$_3$ and 0.1 M NaCl and, 0.81 M HNO$_3$ and 0.1 M NaBrO$_3$, respectively. (**1**) Heating speed = 1.6 °C min$^{-1}$, share rate = 45 s$^{-1}$. (**2**) Heating speed = 1.6 °C min$^{-1}$, $\gamma$ = 2.0%, $f$ = 1.0 Hz. (**3**) The correlation function was collected via 2.0 s laser irradiation.

triblock copolymers, sequential aggregation was observed in the reduced state, that is, the oscillating profile showed sharp top peaks. In contrast, in a previous study using the ABA triblock copolymer, a plateau region appeared in the oscillating profiles in the reduced state[9]. Therefore, we conclude that due to the stepwise aggregation process, the micelle structures of the ABC triblock copolymer synthesized in this study must have fewer loop chains than those of the ABA triblock copolymer. Further, the outermost surface of the ABC triblock copolymer based micelles must be covered with connectable self-oscillating segments[16,17].

Thus, the sequential aggregation behaviour in the reduced state strongly depends on the polymer architecture; that is, ABA or ABC. This direct comparison of the two types of triblock copolymers implies that the dynamic formation of the micelle network structure occurred efficiently in the case of the ABC triblock copolymer.

**Macroscopic rheological oscillation of the block copolymer.** We remark that the oscillation amplitude of $\sigma_I^2$ was not large enough to achieve percolation of the micelle network structure because of the dilute polymer concentration (Supplementary Figs 4c and 5c). However, the oscillation amplitude of $\sigma_I^2$ for a concentrated polymer solution (5.0 wt%) showed a significant increase in the amplitude (Fig. 3a). This result encouraged us to investigate viscosity and viscoelastic oscillation under high concentrations of the ABC triblock copolymer. In this connection, macroscopic viscoelastic oscillation originating from microscopic structural oscillations between association and dissociation of the percolated micellar network structure, similar to that observed in a living organism[19], were expected.

Figure 3b shows changes in the viscosity of ABC triblock copolymer solutions (1.0–5.0 wt%) coupled with the BZ reaction. Autonomous viscosity oscillations were observed for all polymer concentrations. In general, the viscosity and the amplitude of viscosity oscillations increased with an increase in the polymer concentration. A double logarithmic plot of the viscosity (or the amplitude of viscosity, respectively) and the polymer concentrations revealed a linear relationship (Supplementary Fig. 6). However, a discontinuity was observed around 3.0 wt%, which corresponds to polymer overlapping concentration (C*) that could be estimated from Supplementary Fig. 7. In addition, the period of the oscillation was positively correlated with the polymer concentration (Supplementary Fig. 8). This correlation can be explained by an increase in the amount of Ru(bpy)$_3$ with increasing polymer concentration[20]. Remarkably, as seen in

Fig. 3b, when the polymer concentration was 5.0 wt%, the viscosity amplitude reached a maximum of 1,960 mPa s. This value is comparable to the amplitude occurring in living amoeba, and about 10$^3$ times larger than those observed in previous reports[7,8,9]. Supplementary Fig. 9a–e shows the oscillating profiles of the viscosity at several temperatures. From the temperature dependence of the period, the activation energy ($E_a$) can be estimated from an Arrhenius plot. $E_a$ of the reaction was calculated to be 72.4 kJ per mol for 5.0 wt% polymer solution, which is comparable to estimates provided in previous reports[21,22] (Supplementary Fig. 9f).

We further examined the oscillation period as a function of the BZ substrates concentration (Supplementary Figs 10 and 11). The oscillation period ($T$) can be expressed using the following empirical relationship for the concentrated ABC triblock copolymer solution (5.0 wt%):

$$T = 1.50[\text{HNO}_3]^{0.624}[\text{NaBrO}_3]^{-0.515}[\text{MA}]^{-0.955}. \quad (1)$$

Further, the oscillation period for the dilute ABC triblock copolymer solution (0.1 wt%) can be expressed as follows:

$$T = 1.78[\text{HNO}_3]^{-0.785}[\text{NaBrO}_3]^{-0.702}[\text{MA}]^{-0.415}. \quad (2)$$

The oscillation period was negatively correlated with [NaBrO$_3$] and [MA] for both the concentrated and the dilute polymer solutions, whereas the oscillation period was positively correlated with [HNO$_3$] for the concentrated solution. This tendency can be explained by taking into account the well-established Field-Körös-Noyes (FKN) mechanism, which describes the elementary reactions of the BZ reaction[23,24] (Supplementary Discussion and Supplementary Fig. 12), and consistently agrees with other reports on self-oscillating gels and self-oscillating polymer solution systems corresponding to concentrated and dilute systems in this study, respectively[9,25]. The concentration of the BZ substrates also affected the viscosity amplitude (Supplementary Fig. 13), because the redox states of Ru(bpy)$_3$ strongly depends on the concentration of the BZ substrates[23,24]. From these results, it was determined that the best condition to obtain the maximum amplitude in the concentrated solution system were [HNO$_3$] = 0.81 M, [NaBrO$_3$] = 0.1 M, [MA] = 0.04 M, $T$ = 30 °C.

At this optimized condition, $G'$ and $G''$ were measured as a function of time to investigate the profile of the viscoelastic oscillation. As a result, not only autonomous viscoelastic oscillations, but also autonomous sol-gel oscillations (that is, cyclic changes of the magnitude of the correlation between $G'$ and $G''$) were observed (Fig. 3c). Although Boekhoven and co-workers

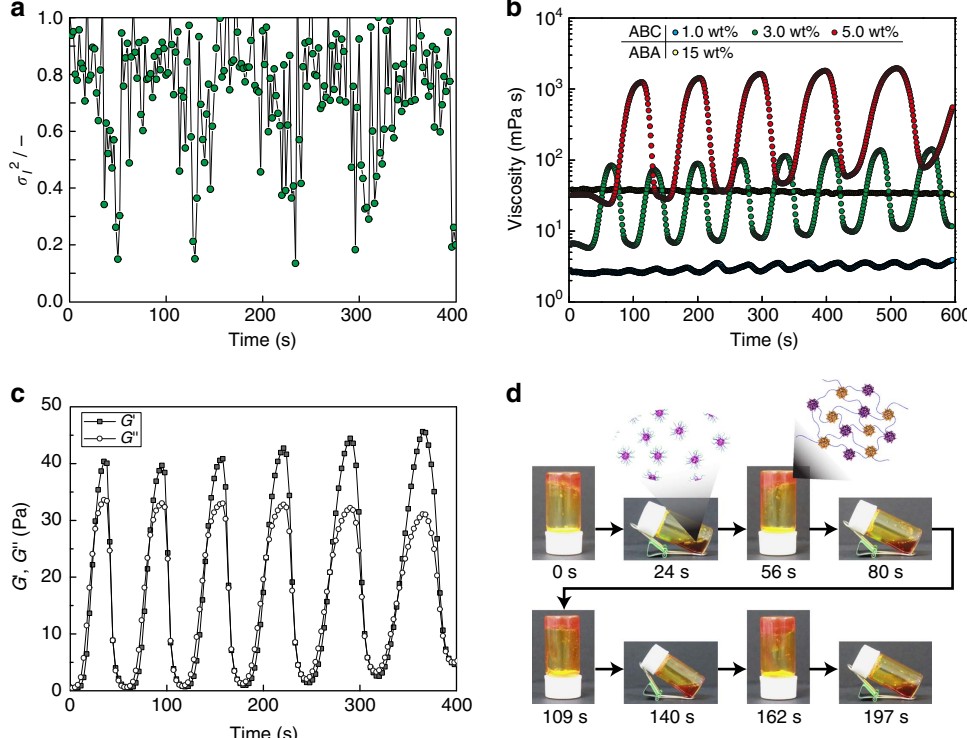

**Figure 3 | Self-oscillating properties of the ABC triblock copolymers.** (**a**) Oscillating behaviours of the initial amplitude of the intensity correlation function, $\sigma_I^2$ during the BZ oscillation reaction for the ABC triblock copolymer solution (5.0 wt%) at 26 °C. The correlation function was collected via 2.0 s laser irradiation. (**b**) Viscosity oscillation of the ABA and ABC triblock copolymer solutions at 26 °C with share rate = 45 s$^{-1}$. (**c**) Oscillating profiles of $G'$ and $G''$ of ABC triblock copolymer solution (5.0 wt%) at 30 °C, with $\gamma$ = 2.0% and $f$ = 1.0 Hz. (**d**) Direct observation of an autonomous sol-gel oscillation of the ABC triblock copolymer solution (5.0 wt%) at 30 °C. These pictures were captured from Supplementary Movie 1. The BZ substrates in the feed for the measurements were composed as follows: [HNO$_3$] = 0.81 M, [NaBrO$_3$] = 0.1 M, [MA] = 0.04 M.

reported synthetic fluids with sol-gel-sol one-way transition accompanied by rearrangement of a low-molecular weight gelation agent, this transient assembly was driven by alkylation of carboxylate (sol to gel) and hydration of carboxylate (gel to sol) after ∼10 h from gelation[26]. Recently, Postma and co-workers reported synthetic fluids with gel-sol-gel one way transition. Even though it was achieved by combining degradation of crosslinking network (gel to sol) and crosslinking reaction of side chain of gel network and crosslinker (sol to gel) after ∼16 h from gelation, it was also not reversible nor repeatable[27]. It is notable that a self-oscillating sol-gel transition was observed repeatedly and reversibly under constant condition without any external stimuli. The oscillating profiles of $G'$ and $G''$ as well as the emergence of the sol-gel oscillation depended on temperature (Supplementary Fig. 14). The self-oscillating sol-gel transition was also directly observed macroscopically as a change in fluidity of the polymer solution (Fig. 3d and Supplementary Movie 1). It was clearly demonstrated that the solution spontaneously and cyclically entered a sol state in the oxidized state and a gel state in the reduced state.

**Autonomous motility oscillation of the block copolymer.**
Figure 4a shows the behaviour of the small droplet of the polymer solution filled inside a tilted glass capillary. We could clearly observe an intermittent forward motion of the droplet inside the capillary. This intermittent motion is due to spontaneous and periodic sol-gel transitions of the polymer solution. The droplet moved forward quickly as a sol because of its increased fluidity during the oxidized state and because of the tilted angle of the glass capillary, whereas the movement of the solution stopped

during the reduced state due to gelation and the accompanying loss of fluidity (Supplementary Movie 2). The stepwise changes in the position and the velocity of the droplet are also shown in Fig. 4b. Thus, we successfully demonstrated an intermittent forward motion of the polymer solution.

## Discussion
In the actual amoeba, such as *Amoeba proteus*, inner plasma of the sol states moves in the travelling direction, and forms gel in front of the pseudopodia. In contrast, outer plasma of the gel states forms sol to replenish the inner plasma of the sol. This periodic sol-gel conversion is responsible for the movement. Here, hydrostatic pressure is generated in the cells by the contraction caused by ATP of actomyosin layer present as the outer plasma. Then, the flow occurs by the relaxation of actomyosin layer at the front with the aid of calcium ions. The membrane potential of amoeba is known to have polarity front and back, which may be involved in the control of the direction of the movement[28]. Also, the propagation of actin waves plays important role to produce deformation[29]. However, it has not been fully elucidated[30-32].

Now, the viscosity of the polymeric fluid is dependent on the molecular weight of the solute, and the elastic modulus of the polymer gel is dependent on the effective crosslinking density. In this study, we successfully demonstrated autonomous sol-gel conversion as a result of the spatio-temporal oscillation of the molecular weight or crosslinking density of the block copolymer solution. This oscillation mechanism is different in molecular level from the actual amoeba. In addition, the polymeric solution showed the intermittent motion by using gravity as the external force to the

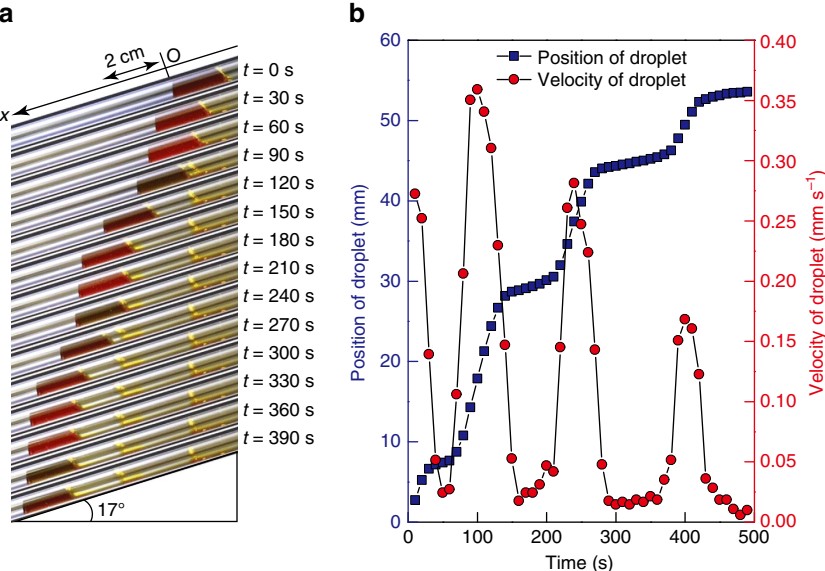

**Figure 4 | Autonomous motility oscillation of the ABC triblock copolymer. (a)** Intermittent forward motion of a droplet of the ABC triblock copolymer solution inside a glass capillary by autonomous and periodic sol-gel transition at 26 °C. **(b)** The position and the velocity of the droplet inside the glass capillary. These results were obtained from Supplementary Movie 2. The BZ substrates in the feed for the measurements were composed as follows: $[HNO_3] = 0.81$ M, $[NaBrO_3] = 0.1$ M, $[MA] = 0.04$ M.

sol-gel oscillation. However, it can be likened to the movement mechanism of the actual amoeba in that both of them use chemical energy as driving force. As future perspective, by utilizing a chemical wave of the BZ reaction, there are possibilities to realize a biomimetic soft-machine which shows amoeboid motion without any external forces such as gravity so that the solution will have polarity front and back as shown in actual amoeba.

In summary, this work demonstrates the amoeba-like intermittent autonomous moving motion under constant condition without any external stimuli by strategically designed synthetic polymers. Self-oscillating sol-gel transition was realized under constant condition without any external stimuli as chemomechanically dynamic bioinspired materials. In the reported mechanism, cyclic association and dissociation of precisely designed synthetic polymers play an important role, similar to cyclic polymerization and depolymerization of actin filaments in living amoeba. In general, research on self-oscillating sol-gel transitions has a strong potential to realize a type of bioinspired autonomous soft robots such as a slime-like robots and soft actuators that can undergo shape transformation kaleidoscopically on demand. In addition, amoeboid motion is observed not only in the movement of amoeba, but also in the development process of multicellular organisms[33], repair of trauma[34], movement of immune cells[35], metastasis of cancer[36] and so on. To clarify these phenomena, our designed ABC triblock copolymer solution can be a theoretical model for analysing the locomotion dynamics of amoeba by simulation studies as constructive approach[37].

## Methods

**Materials.** The chain transfer agent (CTA) S-1-dodecyl-S'-(α,α'-dimethyl-α''-acetic acid) trithiocarbonate was purchased from Trylead Corporation. NIPAAm was kindly supplied from the Kojin Corporation and purified via recrystallizations from a toluene/hexane mixed solvent. N, N-Dimethylacrylamide (DMAAm) and n-butylacrylate (BA) was purchased from Wako Chemicals and purified via vacuum distillation. N-(3-Aminopropyl)methacrylamide hydrochloride (NAPMAm) was purchased from Polyscience and used as received. Bis(2,2'-bipyridine)(1-(4'-methyl-2,2'-bipyridine-4-carbonyloxy)-2,5(pyrrolidine-dione)ruthenium(II) bis(hexafluorophosphate) (Ru(bpy)₃-NHS) was synthesized

according to the previously reported procedure[38]. All other chemical reagents were purchased from Wako Chemicals and used as received unless otherwise noted.

**Synthesis of the self-oscillating ABC triblock copolymer.** Synthesis of the ABC triblock copolymer was accomplished by sequential RAFT random copolymerization[39]. The first step was the preparation of P(NIPAAm-r-BA)-CTA. A round-bottom flask was charged with CTA (200 mg, $5.48 \times 10^{-4}$ mol), NIPAAm (30.0 g, $2.65 \times 10^{-1}$ mol), BA (3.78 g, 4.24 ml, $2.95 \times 10^{-2}$ mol) ([NIPAAm]/[BA] = 90/10 mol%), 2,2'-azobis(2,4-dimethylvaleronitrile) (V65) (2.72 mg, $1.10 \times 10^{-5}$ mol), and 1,4-dioxane (295 ml) and purged with Ar for 30 min. RAFT random copolymerization was carried out at 60 °C for 1.5 h. The product was concentrated under reduced pressure, and it was purified by three rounds of reprecipitation from acetone, a good solvent, and diethyl ether, a poor solvent, and dried in a vacuum oven at room temperature.

Next, a round-bottom flask was charged with P(NIPAAm-r-BA)-CTA (3.5 g, $2.75 \times 10^{-4}$ mol), DMAAm (19.3 g, 20 ml, $1.95 \times 10^{-1}$ mol), V65 (1.37 mg, $5.50 \times 10^{-6}$ mol) and 1,4-dioxane (97.4 ml) and purged with Ar for 30 min. RAFT random copolymerization was carried out at 60 °C for 2 h. The product was concentrated under reduced pressure, and it was purified by three rounds of reprecipitation from acetone, a good solvent and diethyl ether, a poor solvent.

Then, a round-bottom flask was charged with P(NIPAAm-r-BA)-b-PDMAAm-CTA (5.00 g, $9.30 \times 10^{-5}$ mol), NIPAAm (3.00 g, $2.65 \times 10^{-2}$ mol), NAPMAm (249 mg, $1.40 \times 10^{-3}$ mol) ([NIPAAm]/[NAPMAm] = 95/5 mol%), V65 (0.462 mg, $1.86 \times 10^{-6}$ mol) and methanol (13.3 ml) and purged with Ar for 30 min. RAFT random copolymerization was carried out at 60 °C for 5 h. The product was concentrated under reduced pressure and purified by two rounds of reprecipitation from acetone, a good solvent and diethyl ether, a poor solvent. The dodecyl trithiocarbonate end group of P(NIPAAm-r-BA)-b-PDMAAm-b-P(NIPAAm-r-NAPMAm)-CTA was removed according to the following procedure. All obtained ABC triblock copolymers and an appropriate amount of 2,2'-azobis(isobutyronitrile) (AIBN) (458 mg, $2.79 \times 10^{-3}$ mol) were dissolved in 150 ml of ethanol, and the solution was purged with Ar for 30 min. The cleavage reaction was performed at 80 °C for 8 h. The product was concentrated under reduced pressure, and the resulting residue was purified by two rounds of reprecipitation from ethanol, a good solvent, and diethyl ether, a poor solvent, and drying under vacuum at room temperature for 24 h.

Subsequently, the ABC triblock copolymer (5.63 g, $7.58 \times 10^{-5}$ mol) was dissolved in DMSO (30 ml) and added to a 20 ml of DMSO solution containing Ru(bpy)₃-NHS (2.75 g, $2.71 \times 10^{-3}$ mol). Triethylamine (126 µl, $9.04 \times 10^{-4}$ mol) was added to the solution, and the solution was stirred for 24 h at room temperature. The obtained polymer was dialyzed against water and finally lyophilized for recovery.

**Dynamic light scattering measurements.** Sample solutions for DLS measurements were passed through 0.20 µm filters to eliminate dust before use. DLS measurements were performed on a DLS/SLS-5,000 compact goniometer

(ALV, Langen, Germany) coupled with an ALV photon correlator. A 22 mW He-Ne laser (Uniphase Co. Ltd., USA) was used as light source. The wavelength of the laser light in vacuum was 632.8 nm. Although the laser power was relatively weak, the output photon count rate was ~50 times higher than that of a conventional pinhole system; this increased photon count rate was achieved by employing a set of static and dynamic enhancers (that is, devices to enhance the photon counting rate) and a high quantum efficiency avalanche photodiode detection system. Experiments were performed at a range of temperatures (10–35 °C) with an accuracy of ±0.1 °C. The intensity autocorrelation functions, $g_2(q, t)$, were recorded at a scattering angle of 90° at each temperature for 2 s or 30 s. Samples were equilibrated at a constant temperature for at least 30 min before data collection. When studying the BZ oscillation reaction, both autocorrelation function and time average scattering intensities were collected at a scattering angle of 90° for every 2 s. Experiments were performed at temperatures from 20 °C to 30 °C with an accuracy of ±0.1 °C. Samples were equilibrated at a constant temperature for at least 30 min before data collection.

For solutions containing monodisperse particles, the electric field correlation function, $g_1(q, t)$, displays a single exponential decay, as follows:

$$g_1(q,t) = \exp(-\Gamma t) = \exp(-D_0 q^2 t), \tag{3}$$

where $q$ is the scattering vector ($q = (4\pi n/\lambda)\sin(\theta/2)$, where $n$ is the refractive index of the solution, $\lambda$ is the wavelength of light in vacuum, and $\theta$ is the scattering angle), $\Gamma$ is the decay rate, and $D_0$ is the translational diffusion coefficient at the infinitely dilute limit. The recorded intensity correlation function, $g_2(q, t)$, was converted to $g_1(q, t)$ using the Siegert relation[40]. The hydrodynamic radius, $R_h$, can be estimated with knowledge of the solvent viscosity, $\eta$, using the Stokes-Einstein equation:

$$R_h = (k_B T)/6\pi\eta D_0. \tag{4}$$

For solutions that contain polydisperse particles, $g_1(q, t)$ can be determined using the method of cumulants, as follows[41]:

$$g_1(q,t) = A\exp(-\Gamma t)(1 + (1/2!)\mu_2 t^2 - (1/3!)\mu_3 t^3), \tag{5}$$

where $\Gamma$ is the mean decay rate and $\mu_2/\Gamma^2$ characterizes the width of the distribution. In this work, the apparent $R_h$ was calculated using equation (4) by substituting $D = \Gamma/q^2$ of a 0.1 wt% solutions for $D_0$.

The reciprocal of $\Gamma$ is defined as relaxation time. The distribution of $\Gamma^{-1}$ is expressed as relaxation time distribution function, $G(\Gamma^{-1})$. It was examined by applying the inverse Laplace transformation to $g_1(q, t)$ using the well-established CONTIN program[42,43].

**Rheological measurements.** Rheological measurements were performed with an Anton Paar Physica MCR 301 rheometer using the parallel plate geometry with 25 mm diameter plates (PP25, $d = 1.0$ mm).

**Data availability.** All data are available from the authors upon reasonable request.

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

## Acknowledgements

This work was supported in part by the Grants-in-Aid for Scientific Research (No. 26620164 and No. 15H05495 awarded to T.U. and No. 15H02198 awarded to R.Y.) from the Ministry of Education, Culture, Sports, Science and Technology of Japan and Research Fellowship of the Japan Society for the Promotion of Science for Young Scientists (No. 14J02019 to R.T. and No. 16J09350 to M.O.).

## Author contributions

M.O. performed all experiments and authored the manuscript. T.U., R.T., M.S. and R.Y. conceived and directed the study, as well as authored the manuscript.

## Additional information

**Competing interests:** The authors declare no competing financial interests.

