## [Peer Review File · Nature Communications]

Reviewers' comments:

Reviewer #1 (Remarks to the Author):

The paper findings are interesting for scientific communities across disciplines, like chemistry, biology, and robotics. The claims are in fact strong, in view of achieving a system moving autonomously, with no need for external stimuli for triggering motion. This is of interest for robotics, for instance. At the same time, the authors claim bioinspiration, i.e. inspiration to the amoeba mechanism of motion, so that their work can be of interest in the field of biology, too. The results are however mainly important for chemistry, as the authors made a progress in their achievements on self-oscillating gels, by largely increasing the range of viscosity oscillation and by making the oscillation autonomous by a sol-gel transition.

The overall claim of autonomous motion is interesting and novel. It can open up interesting perspectives for autonomous systems, especially at the microscale, likely in the biomedical field.

The authors do not elaborate on the aspect of the possible applications of their results.

The paper strongly relies on previous achievements by the same authors. This gives solidity to the work and findings. On this regard, 14 publications by the authors (out of 26 total references) are cited, by explicitly signaling them as theirs. The progress that the authors outline with respect to previous achievements are the range of viscosity oscillation and the independence from external triggering.

Concerning the presentation of the paper, the claims are very clear and strong. The methods and results are well presented, with good level of details. The actual principle that brings to the improvement of the self-oscillating mechanism could have been elaborated more. The paper goes from the claims to the description of the work. Especially in view of an interdisciplinary readership, between the claims and the description, the principles could be explained more precisely, especially for outlining the novel ideas.

The inspiration from the amoeba mechanism of locomotion is very interesting and quite well investigated for robotics applications. The paper however just mentions this bioinspiration in the introduction and in the conclusions. It would be interesting to outline and elaborate on the precise principles and mechanisms that have been adopted and to what level.

The videos do not convey their message very clearly. Some text could probably help to describe what the videos are intended to show and how the set-ups have been chosen. For instance the 17 deg of inclination of the capillary in the second video, which conveys some gravity contribution. The work is interesting and the results are valuable, with a good progress with respect to the solid background they start from. The paper can be improved in presentation, to discuss more aspects that are claimed.

Reviewer #2 (Remarks to the Author):

This is a novel work in which the redox change of $\text{Ru}(\text{bpy})_3$ driven by the BZ reaction causes very large viscosity changes in a solution of an ABC triblock copolymer. The work is excellent, and the paper is well written. This is an exciting work that will have a significant influence in the field of soft matter and adaptive materials. I recommend publication.

Reviewer #3 (Remarks to the Author):

This manuscript is very interesting and is likely to attract attention. Demonstrating the ability of an artificial material to rather abruptly start and stop its flow due to its cyclic modification by interacting with a chemical oscillator (BZ) reaction is a feat. I recommend publication, with two caveats.

1. It is going too far to call the present material an artificial amoeba. Although sol-gel transitions

are part of an amoeba's motion (and also motions of many cell types), biological motility is much more complex. An external force such as gravity was needed to move the present artificial material. I suggest dropping "Artificial amoeba" from the title and removing the emphasis on amoeba from the introduction. In my opinion, "Self-oscillating polymeric fluid with abrupt sol-gel and motility transition" would be a more accurate title.

2. The equations relating period of oscillation T to chemical composition have not really been demonstrated. All that has been shown is that each component, behaves as a power law when the other component concentrations are held constant. You would need to show that you could accurately predict T when simultaneously varying two or three of the components.

Is there a reason why the exponent for $[\text{HNO}_3]$ is positive for 5 wt% polymer but negative for 0.1 wt% polymer?

Responses to Reviewer #1

(Reviewer's comments 1/5 (general comment))

The paper findings are interesting for scientific communities across disciplines, like chemistry, biology, and robotics. The claims are in fact strong, in view of achieving a system moving autonomously, with no need for external stimuli for triggering motion. This is of interest for robotics, for instance. At the same time, the authors claim bioinspiration, i.e. inspiration to the amoeba mechanism of motion, so that their work can be of interest in the field of biology, too. The results are however mainly important for chemistry, as the authors made a progress in their achievements on self-oscillating gels, by largely increasing the range of viscosity oscillation and by making the oscillation autonomous by a sol-gel transition.

The overall claim of autonomous motion is interesting and novel. It can open up interesting perspectives for autonomous systems, especially at the microscale, likely in the biomedical field.

(Answer for 1/5)

We would greatly appreciate a high evaluation of our paper and your helpful comments to make a revised manuscript. We have revised the manuscript by taking all of your comments into consideration. Answers to your comments are as follows:

(Reviewer's comments 2/5)

The authors do not elaborate on the aspect of the possible applications of their results.

(Answer for 2/5)

As a possible application, our study has a strong potential for slime-like robots, soft actuators, or the theoretical model for analyzing the locomotion dynamics of amoeba. From this point of view, we added the sentences about the possible applications and future perspectives of our results in the last paragraph as follows:

In addition, amoeboid motion is observed not only in the movement of amoeba, but also in the development process of multicellular organisms, repair of trauma, movement of immune cells, metastasis of cancer, etc. To clarify these phenomena, our designed ABC polymer solution can be a theoretical model for analyzing the locomotion dynamics of amoeba by simulation studies as constructive approach.

(Reviewer's comments 3/5)

The paper strongly relies on previous achievements by the same authors. This gives solidity to the work and findings. On this regard, 14 publications by the authors (out of 26 total references) are cited, by explicitly signaling them as theirs. The progress that the authors outline with respect to previous achievements are the range of viscosity oscillation and the independence from external triggering.

Concerning the presentation of the paper, the claims are very clear and strong. The methods and results are well presented, with good level of details. The actual principle that brings to the improvement of the self-oscillating mechanism could have been elaborated more. The paper goes from the claims to the description of the work. Especially in view of an interdisciplinary readership, between the claims and the description, the principles could be explained more precisely, especially for outlining the novel ideas.

(Answer for 3/5)

Regarding to the comment on the necessity for the design principle of the viscoelasticity oscillation, we added short description on it at the end of the last paragraph of the introduction. Our basic idea to yield G' (elastic modulus) oscillation is to provoke periodic changes of the number of effective polymer junction from which polymer strands emanate in space driven by the BZ reaction. This is based on the well-established network percolation theory that confirms G' is proportional to the number density of the cross-linking point. When the C segment is oxidized, block copolymer micelles are simply solubilized results in the viscous solution. Conversely, effective cross-links are formed and percolated into the space to provide gel state, when C is reduced during the BZ reaction.

(Reviewer's comments 4/5)

The inspiration from the amoeba mechanism of locomotion is very interesting and quite well investigated for robotics applications. The paper however just mentions this bioinspiration in the introduction and in the conclusions. It would be interesting to outline and elaborate on the precise principles and mechanisms that have been adopted and to what level.

(Answer for 4/5)

We added the sentences about the relationship between the dynamics of live amoeba and the dynamics of our artificial amoeba in the last subheadings section (**Autonomous motility oscillation of the block copolymer**) as follows:

In the actual amoeba, such as *Amoeba proteus*, inner plasma of the sol states moves in the traveling direction, and forms gel in the front of the pseudopodia. In contrast, outer plasma of the gel states forms sol to replenish the inner plasma of the sol. This periodic sol-gel conversion is responsible for the movement. Here, hydrostatic pressure is generated in the cells by the contraction caused by ATP of actomyosin layer present as the outer plasma. Then, the flow occurs by the relaxation of actomyosin layer at the front with the aid of calcium ion. The membrane potential of amoeba are known to have polarity front and back, which may be involved in the control of the direction of the movement. Also, the propagation of actin waves plays important role to produce deformation. However, it has not been fully elucidated.

Now, the viscosity of the polymeric fluid is dependent on the molecular weight of the solute, and the elastic modulus of the polymer gel is dependent on the effective crosslinking density. In this study, we successfully demonstrated autonomous sol-gel conversion as a result of the spatio-temporal oscillation of the molecular weight or crosslinking density of the block copolymer solution. This oscillation mechanism is different in molecular level from the actual amoeba. In addition, our artificial amoeba solution showed the amoeboid motion by using

gravity as the external force to the sol-gel oscillation. However, it can be likened to the movement mechanism of the actual amoeba in that both of them use chemical energy as driving force. As future perspective, by utilizing a chemical wave of BZ reaction, there are possibilities to realize a biomimetic soft-machine which shows amoeboid motion without any external forces such as gravity so that the solution will have polarity front and back as shown in actual amoeba.

(Reviewer's comments 5/5)

The videos do not convey their message very clearly. Some text could probably help to describe aht the videos are intended to show and how the set-ups have been chosen. For instance the 17 deg of inclination of the capillary in the second video, which conveys some gravity contribution.

(Answer for 5/5)

We added the sentences as follows in the main text:

(see supplementary information for details on experimental setup of the movie)

Also we added the sentences about the detailed experimental setup of the movie in the supplementary information as follows:

In order to clearly observe the sol-gel oscillation as a change in motility, the polymer solution and the substrate were mixed and immediately enclosed in the capillary.

Responses to Reviewer #2

(Reviewer's comment 1/1 (general comment))

This is a novel work in which the redox change of Ru(bpy)₃ driven by the BZ reaction causes very large viscosity changes in a solution of an ABC triblock copolymer. The work is excellent, and the paper is well written. This is an exciting work that will have a significant influence in the field of soft matter and adaptive materials. I recommend publication.

(Answer for 1/1)

We would greatly appreciate a high evaluation of our original paper. We have revised the manuscript keeping the context and story of the manuscript unchanged. According to the comments from other reviewers, we revised the manuscript from following aspects: **1.** Detail description of experimental set-ups, **2.** In what level the sol-gel oscillation could be adopted as reproduction of amoeba dynamics, and **3.** Possible applications and future perspectives.

Responses to Reviewer #3

(Reviewer's comments 1/4 (general comment))

This manuscript is very interesting and is likely to attract attention. Demonstrating the ability of an artificial material to rather abruptly start and stop its flow due to its cyclic modification by interacting with a chemical oscillator (BZ) reaction is a feat. I recommend publication, with two caveats.

We would greatly appreciate a high evaluation of our paper and your helpful comments to make a revised manuscript. We have revised the manuscript by taking all of your comments into consideration. Answers to your comments are as follows:

(Reviewer's comments 2/4)

1. It is going too far to call the present material an artificial amoeba. Although sol-gel transitions are part of an amoeba's motion (and also motions of many cell types), biological motility is much more complex. An external force such as gravity was needed to move the present artificial material. I suggest dropping "Artificial amoeba" from the title and removing the emphasis on amoeba from the introduction. In my opinion, "Self-oscillating polymeric fluid with abrupt sol-gel and motility transition" would be a more accurate title.

(Answer for 2/4)

As you pointed out, the present material showed just the autonomous sol-gel oscillation, and it was hard to observe unidirectional movement of the polymeric fluids without the help of the gravity. However, we still believe that there is a strong potential evolving to soft-robotics materials which can move on the flat surface.

In live amoeba, it is well known that wave propagation as well as sol-gel oscillation is highly important to produce movement^[1]. As we revised in the last subheading section (Autonomous motility oscillation of the block copolymer), one of the important feature of our materials is that there is a wave propagation (polarity) which can be directly translated as the sol-gel propagation. During the wave propagation of BZ reaction, not only motility, but also surface tension of the droplet is propagating. In previous article, this surface tension itself could produce the movement of BZ droplet in oil^[2]. These features and fact strongly reflect not only the application potential to create soft-robotics materials which can move on the flat surface, but also the academic potential to simplify the movement of amoeba as an active matter in the field of biophysics by using our chemically synthesized polymeric fluids.

Thus, we concluded that it is not too far to call the present material an artificial amoeba because of the versatile potential mentioned above.

[1] Phase geometries of two-dimensional excitable waves govern self-organized morphodynamics of amoeboid cells, *PNAS*, **110**, 13 (2013).

[2] Spontaneous motion of a droplet coupled with a chemical wave, *Physical Review E* **84**, 015101(R) (2011).

(Reviewer's comments 3/4)

2. The equations relating period of oscillation T to chemical composition have not really been demonstrated. All that has been shown is that each component, behaves as a power law when the other component concentrations are held constant. You would need to show that you could accurately predict T when simultaneously varying two or three of the components.

(Answer for 3/4)

We newly added **Figure S11** in the revised Supplementary information. As shown in **Figure S11**, each result in **Supplementary Fig. S10** can also be expressed by a linear line. Therefore, we could describe the equations relating period of oscillation T to chemical composition as follows^[2]:

$$T = 1.78[\text{HNO}_3]^{-0.785}[\text{NaBrO}_3]^{-0.702}[\text{MA}]^{-0.415}; ([\text{Polymer}] = 0.1 \text{ wt}\%)$$

$$T = 1.50[\text{HNO}_3]^{0.624}[\text{NaBrO}_3]^{-0.515}[\text{MA}]^{-0.955}; ([\text{Polymer}] = 5.0 \text{ wt}\%)$$

Figure S11 Period as a function of the initial concentrations of HNO₃, NaBrO₃, and MA for (a) the 0.1 wt% solution and (b) the 5.0 wt% solution.

[2] Yoshida, R., Onodera, S., Yamaguchi, T. & Kokufuta, E. Aspects of the Belousov–Zhabotinsky reaction in polymer gels. *J. Phys. Chem.* **103**, 8573–8578 (1999)

(Reviewer's comments 4/4)

Is there a reason why the exponent for [HNO₃] is positive for 5 wt% polymer but negative for 0.1 wt% polymer?

(Answer 4/4)

We compare the waveforms of the oscillation in each concentration condition to elucidate the factors affecting opposite dependency on the [HNO₃] depending on the [Polymer]. We newly added **Figure S12**, **Table S2**, **S3**, and **S4** in revised supplementary information. FKN mechanism could be divided into three sub-processes, i.e., process A (consumption of bromide ion), process B (oxidation of Ru(bpy)₃²⁺, autocatalytic reaction), and process C (reduction of Ru(bpy)₃³⁺, production of bromomalonic acid) as shown below;

FKN mechanism

Process A (consumption of bromide ion)

Process B (oxidation of Ru(bpy)₃²⁺, autocatalytic reaction)

Process C (reduction of Ru(bpy)₃³⁺, production of bromomalonic acid)

By analyzing waveform of BZ reaction (**Figure S12**), we could trace each process in detail (**Table S3**, **Table S4**):

Figure S12 Waveform analysis of ABC triblock copolymer solution at 26 °C. The BZ substrates in the feeds for the measurements were summarized in **Table S2**.

Table S2 The BZ substrates in the feeds for the measurements of **Figure S12**.

Figure	[Polymer] / wt%	[HNO ₃] / M	[NaBrO ₃] / M	[MA] / M
(a)	0.1	0.60	0.1	0.04
(b)	0.1	0.81	0.1	0.04
(c)	5.0	0.70	0.1	0.04
(d)	5.0	0.90	0.1	0.04

Table S3 Periods of each process in the course of BZ reaction of the 0.1 wt% polymer solution of **Figure S12**.

[HNO ₃] / M	Process A / s	Process B / s	Process C / s	Total Period / s
0.60	17	17	32	66
0.81	5	19	29	53

Table S4 Periods of each process in the course of BZ reaction of the 5.0 wt% polymer solution of **Figure S12**.

[HNO ₃] / M	Process A / s	Process B / s	Process C / s	Total Period / s
0.70	9	44	32	85
0.90	3	44	43	90

Here, for simplicity, we assume that reaction kinetics of process B involving autocatalytic reaction characterizing BZ reaction is not changed by changes in the BZ reaction condition. In 0.1 wt% polymer solution, process A was accelerated while the duration of process C was not changed so much by increasing [HNO₃] from 0.60 M to 0.81 M^[3]. As a result, a total oscillation period became short. In contrast, in 5.0 wt% polymer solution, the duration

of process A became shorter, whereas that of process C did longer with increases in $[\text{HNO}_3]$ from 0.70 M to 0.90 M. Therefore, the difference in the H^+ dependency on the process C is essential.

Note that the total amount of $[\text{Ru}(\text{bpy})_3]$ in the solution linearly depends on the $[\text{Polymer}]$.

In 0.1 wt% polymer solution, $[\text{Ru}(\text{bpy})_3]$ was diluted and $[\text{H}^+]$ was saturated against $[\text{Ru}(\text{bpy})_3]$. In other words, reaction kinetics of process C is determined by the $[\text{Ru}(\text{bpy})_3]$ introduced. This is the reason why process C was almost constant even if the $[\text{H}^+]$ increased. On the other hand, process A, in which $\text{Ru}(\text{bpy})_3$ does not participate the reaction, was consistently accelerated with increases in $[\text{H}^+]$ in terms of chemical equilibrium (H^+ , BrO_3^- , and other reagents in solution). Consequently, the total period governed by process A being shorter with increases in $[\text{HNO}_3]$.

In 5.0 wt% polymer solution, we have to close look at the duration of the process C occupied in a period because $[\text{Ru}(\text{bpy})_3]$ was no longer diluted against $[\text{H}^+]$ and sufficient amount of $[\text{Ru}(\text{bpy})_3]$ seemed to be supplied. In this case, H^+ is involved in equation (7)-(10) in the right-hand side in process C, thus the kinetics of these reactions must be reduced. As a result, process C was significantly shortened with increases in $[\text{HNO}_3]$. In summary, the period was increased with increases in $[\text{HNO}_3]$ because the deceleration of process C was dominant rather than the acceleration of process A.

We added these discussions in main text briefly and in supplementary information in detail.

[3] Suzuki, D., Yoshida, R. Effect of initial substrate concentration of the Belousov-Zhabotinsky reaction on self-oscillation for microgel system, *J. Phys. Chem. B.* **112**(40), 12618-12624 (2008).

REVIEWERS' COMMENTS:

Reviewer #1 (Remarks to the Author):

The revised paper is very good and presents nicely the interesting work. The comments received from the reviewers have been taken into account, though not in a very capillary way, and the paper has been generally improved.

Minor comments: few references seem to have been added, but this is not explained in the rebuttal; the videos could helpfully contain some text, in the video itself, that explains what is happening, along with the movie run.

Reviewer #3 (Remarks to the Author):

As indicated in the original review cycle, I think this is extremely interesting and important worksuitable for publication in Nature Communications. However, I still object to the term "Artificial Amoeba." While I understand the value of synthetic biomimesis, and that temporal self-assembly and reversible sol-gel transitions are inherent to cell motility, the present work does not go far enough to mimic the kind of purposeful motion of even such a primitive organism as the amoeba. The gravity driven start-stop action shown in Movie 2 is more reminiscent of a brake or clutch system in an automobile that periodically catches and releases on a steep hill, than it is of a moving cell.

A few other minor notes:

1. The authors use the term "responsibility" more than once in the Introduction, when I believe they mean "responsiveness."
2. I suggest that reference to the TCA cycle be taken out. Although the BZ reaction was originally conceived as a model of TCA, the latter does not run as a coordinated cycle as does BZ.
3. Please elaborate on the meaning of σ_I^2 . I understand it to be that variance of scattered light intensity. Why is it unitless? It is also not clear why percolation is not inferred to occur. One needs a percolating network to achieve a G' exceeding G'' , and that certainly is observed in Figs 2b and 3c.

Responses to Reviewer #1

(Reviewer's comments 1/2)

The revised paper is very good and presents nicely the interesting work. The comments received from the reviewers have been taken into account, though not in a very capillary way, and the paper has been generally improved.

(Answer for 1/2)

We would greatly appreciate a high evaluation of our paper and your helpful comments to make a revised manuscript. We have revised the manuscript by taking all of your comments into consideration. Answers to your comments are as follow:

(Reviewer's comments 2/2)

Minor comments: few references seem to have been added, but this is not explained in the rebuttal;

(Answer for 2/2)

We revised **macroscopic rheological oscillation of the block copolymer** section to add the explanations and rebuttal to the references as follows:

As a result, not only autonomous viscoelastic oscillations, but also autonomous sol-gel oscillations (i.e., cyclic changes of the magnitude of the correlation between G' and G'') were observed (**Fig. 3c**). Although Boekhoven and co-workers reported synthetic fluids with sol-gel-sol one-way transition accompanied by rearrangement of a low molecular weight gelation agent, this transient assembly was driven by alkylation of carboxylate (sol to gel) and hydration of carboxylate (gel to sol) after ~ 10 hours from gelation²⁶. Recently, Postma and co-workers reported synthetic fluids with gel-sol-gel one way transition. Even though it was achieved by combining degradation of crosslinking network (gel to sol) and crosslinking reaction of side chain of gel network and crosslinker (sol to gel) after ~ 16 hours from gelation, it was also not reversible nor repeatable²⁷. It is notable that a self-oscillating sol-gel transition was observed repeatedly and reversibly under constant condition without any external stimuli.

Instead, summary paragraph was slightly simplified.

(Reviewer's comments 2/2)

the videos could helpfully contain some text, in the video itself, that explains what is happening, along with the movie run.

(Answer for 2/2)

We re-edited the caption of the movie to include brief explanation to help easier comprehension for readers as follows:

Supplementary Movie 1 Observation of an autonomous sol-gel oscillation of an ABC triblock copolymer solution (5.0 wt%) at 30 °C. The fluidity of the solution was clearly oscillated repeatedly under constant condition coupled with the redox change of the Ru(bpy)₃ during the BZ reaction. The BZ substrates in the feed for the measurements were composed as follows: [HNO₃] = 0.81 M, [NaBrO₃] = 0.1 M, [MA] = 0.04 M. The movie speed is 3× actual speed.

Responses to Reviewer #3

(Reviewer's comments 1/4 (general comment))

As indicated in the original review cycle, I think this is extremely interesting and important work suitable for publication in Nature Communications. However, I still object to the term "Artificial Amoeba." While I understand the value of synthetic biomimesis, and that temporal self-assembly and reversible sol-gel transitions are inherent to cell motility, the present work does not go far enough to mimic the kind of purposeful motion of even such a primitive organism as the amoeba. The gravity driven start-stop action shown in Movie 2 is more reminiscent of a brake or clutch system in an automobile that periodically catches and releases on a steep hill, than it is of a moving cell.

(Answer for 1/4)

We would greatly appreciate a high evaluation of our paper and your helpful comments to make a revised manuscript. We have revised the manuscript by taking all of your comments into consideration, especially in terms of the term "Artificial Amoeba". The term was deleted from the title, and the claims were toned down in the main text.

(Reviewer's comments 2/4)

A few other minor notes:

1. The authors use the term "responsibility" more than once in the Introduction, when I believe they mean "responsiveness."

(Answer for 2/4)

We replaced the word "responsibility" to "responsiveness" in the introduction. Thank you very much for the comment.

(Reviewer's comments 3/4)

2. I suggest that reference to the TCA cycle be taken out. Although the BZ reaction was originally conceived as a model of TCA, the latter does not run as a coordinated cycle as does BZ.

(Answer for 3/4)

The reference to the TCA cycle was taken out from the main text.

(Reviewer's comments 4/4)

3. Please elaborate on the meaning of σ_I^2 . I understand it to be that variance of scattered light intensity. Why is it unitless? It is also not clear why percolation is not inferred to occur. One needs a percolating network to achieve a G' exceeding G'' , and that certainly is observed in Figs 2b and 3c.

(Answer for 4/4)

In the field of light scattering, σ_I^2 is generally described as follows:

$$\sigma_I^2 + 1 \equiv \lim_{\tau \rightarrow 0} \frac{\langle I_p(t) I_p(t + \tau) \rangle}{\langle I_p(t) \rangle^2} \quad (1)$$

where $I(t)$ is the light scattering intensity as a function of time.

As shown in the right side of the equation (1), it should be unitless.

Also, σ_I^2 is an indicator which represents ergodicity of samples, not directly represents network percolation of samples. In brief, when time-averaged σ_I^2 is equal to ensemble-averaged σ_I^2 , the system is ergodic whereas it is not equal to ensemble-averaged σ_I^2 , the system is non-ergodic^{[1], [2]}. The value of σ_I^2 for ideal ergodic or non-ergodic media can be mathematically calculated and determined as follows:

	Time-averaged σ_I^2	Ensemble-averaged σ_I^2
Ergodic	1	1
Non-ergodic	0	1

When the samples are partially suppressed to diffuse, such as the case of partially crosslinked physical gel, the system can be considered as superposition of ergodic and non-ergodic, and time-averaged σ_I^2 vary from 0 to 1. Thus, time-averaged σ_I^2 can be used as the indicator of the ergodicity of samples (experimentally, initial amplitude of time-intensity correlation function is comparable to time-averaged σ_I^2). It is also known that the rheological gelation temperature (crossover point of G' and G'') of gelatin solution is correspond with inflection point of time-averaged σ_I^2 ^[3].

In our paper, we used time-averaged σ_I^2 as an indicator of the ergodicity of the polymeric solution. By evaluating not only G' and G'' as a macroscopic view point, but also time-averaged σ_I^2 as a microscopic view point, we aim to access the property of the solution more comprehensively. Please note that it is not directly represents gelation point or percolation.

[1] Pusey, P. N. and Megen, W. V. Dynamic Light Scattering by Non-Ergodic Media. *Physica A*, **157**, 705-741 (1989).

[2] Shibayama, M., Norisuye, T. Gel Formation Analyses by Dynamic Light Scattering. *Bull. Chem. Soc Jpn.*, **75**, 647-659 (2002).

[3] Matsunaga, T., Shibayama, M. Gel point determination of gelatin hydrogels by dynamic light scattering and rheological measurements. *Physical Review E*, **76**, 030401(R) (2007).